# A Quantitative and Qualitative Clinical Validation of Soft Tissue Simulation for Orthognathic Surgery Planning

**DOI:** 10.3390/jpm12091460

**Published:** 2022-09-06

**Authors:** Alessandro Gutiérrez Venturini, Jorge Guiñales Díaz de Cevallos, José Luis del Castillo Pardo de Vera, Patricia Alcañiz Aladrén, Carlos Illana Alejandro, José Luis Cebrián Carretero

**Affiliations:** 1Fundación Para La Investigación Biomédica del Hospital Universitario La Paz, 28029 Madrid, Spain; 2Hospital Universitario La Paz, 28046 Madrid, Spain; 3GMV Innovating Solutions, 28760 Madrid, Spain

**Keywords:** orthognathic surgery, validation study, soft tissue simulation, computer-aided surgery, finite element analysis

## Abstract

The purpose of this study was to perform a quantitative and qualitative validation of a soft tissue simulation pipeline for orthognathic surgery planning, necessary for clinical use. Simulation results were retrospectively obtained in 10 patients who underwent orthognathic surgery. Quantitatively, error was measured at 9 anatomical landmarks for each patient and different types of comparative analysis were performed considering two mesh resolutions, clinically accepted error, simulation time and error measured by means of percentage of the whole surface. Qualitatively, evaluation and binary questions were asked to two surgeons, both before and after seeing the actual surgical outcome, and their answers were compared. Finally, the quantitative and qualitative results were compared to check if these two types of validation are correlated. The quantitative results were accurate, with greater errors corresponding to gonions and lower lip. Qualitatively, surgeons answered similarly mostly and their evaluations improved when seeing the actual outcome of the surgery. The quantitative validation was not correlated to the qualitative validation. In this study, quantitative and qualitative validations were performed and compared, and the need to carry out both types of analysis in validation studies of soft tissue simulation software for orthognathic surgery planning was proved.

## 1. Introduction

Orthognathic surgery seeks to restore the balance of functionality and aesthetics of the face, specifically of bones, teeth and soft tissue. In the era of virtual planning, different software tools have been developed [1,2] to help both the surgeon and the patient, who is increasingly important in the planning process [3]. These advanced technologies combine volumes (for example cone-beam computed tomography or CBCT for volumetric medical imaging) and surfaces (for example laser scanners for surface reconstruction) through multimodal registration (by means of points, surfaces or voxels) and are useful in preoperative planning as well as in postoperative analysis [4]. Nowadays, the biggest challenge of these programs is the correct prediction of the deformation of the facial soft tissue after the surgery, which is highly dependent on the surgical procedure and the patient characteristics.

In order to be valid for clinical use, a soft tissue simulation software should search a compromise between accuracy and computation time, which means simulations within a clinically accepted error and an interactive interface for the surgeon. In addition, the role of skin texture in the communication with patients is highly important since it allows them to recognize themselves and make them more receptive to the proposed surgery [4]. In summary, a valid surgical planning software for orthognathic surgery must be accurate, fast and realistic. Moreover, before clinical practice, the program must undergo a validation process, to ensure that the results are reliable.

Among the most advanced solutions, both commercial and research based, that meet those requirements, only a few have passed detailed validation confirming that the results are clinically correct. In a recent review of the literature [5], the authors summarize these validations, which are mainly quantitative, and stress the need for qualitative assessments by surgeons. On the one hand, a quantitative measure of the simulation error with respect to the real surgical result, which must be smaller than the clinically accepted error, is necessary. If an average error of the entire face is taken, the result is not reliable, since some regions that do not change after the surgery may underestimate the error [6]. Therefore, this measurement must be made considering the key regions in the planning of orthognathic surgery, namely nose, lips and chin. To do this, simulation results can be evaluated by means of anatomical points of interest or by regions defined with different landmarks and planes [7,8]. On the other hand, qualitatively, the opinion of the surgeons on the reliability of the results is necessary [9]. For this purpose, several questions concerning the different aspects of the prediction software can be defined and asked to the surgeons participating in the study.

Very few studies [10,11,12,13,14] present both quantitative and qualitative validation. Moreover, no publication to date has studied the correlation between them, in order to demonstrate that both are necessary for a complete clinical validation. In a recent study [15], a new soft tissue simulation pipeline based on a finite element model (FEM) was detailed and tested with a cohort of 10 patients, where the percentage of each resulting mesh below the clinically accepted error in orthognathic surgery planning considered by the authors (3 mm) was analysed. In that publication, the aforementioned requirements of accuracy, speed and realism were included, but only a first quantitative validation of the results for the whole face was carried out.

The aim of this study is to perform a quantitative and qualitative clinical validation of the soft tissue simulation pipeline presented in Alcañiz et al. [15]. Both types of validation were compared to determine whether they are correlated or not, and therefore whether they are both necessary.

## 2. Materials and Methods

For this study, the same cohort of 10 patients presented in the study of Alcañiz et al. [15] was selected. The inclusion criteria were: patient had a diagnosis of maxillary deformities and underwent orthognathic surgery at Hospital Universitario La Paz (Madrid, Spain), and both pre and post operative CBCT images are available. The following data were collected: gender (8 women and 2 men), age (mean 32 years, range 22–51 years), ethnic group (8 Caucasian and 2 Latin American), and diagnosis (2 class II malocclusion cases, 4 class III malocclusion cases, 3 asymmetry cases and 1 open bite case). The surgeries exhibit diverse procedures for both the maxilla and the mandible, which allows ample testing of the proposed simulation methodology. Patient characteristics and surgical procedures are detailed in Alcañiz et al. [15]. Approval for the study was obtained from the Ethics Committee of Hospital Universitario La Paz (with protocol code “HULP PI-3755”, on 12 September 2019), as well as informed consent from all the participating patients.

Preoperative and postoperative CBCT images and three-dimensional (3D) scans capture, registration and segmentation, meshes preparation, material properties and boundary conditions definition, and soft tissue simulation and subsequent texturing are described in that previous work [15]. In Figure 1, a scheme of the entire workflow is presented, from the patient data collection to the final validation performed in that study.

As stated in the introduction, in this study, first a quantitative validation was performed and then a qualitative validation was carried out. Both validations were compared afterwards.

### 2.1. Quantitative Validation

In order to know the simulation error in the most critical regions for the surgical planning, the distance (in mm) between the simulation results and the actual surgical outcomes for the 10 patients was taken in the following 9 anatomical landmarks, as requested by an orthognathic surgeon: Subnasale (Sn), soft tissue A point (A), Upper Lip (UL), Lower Lip (LL), soft tissue B point (B), Pogonion (Pog), Menton (Me), Right Gonion (RGo) and Left Gonion (LGo).

Those measurements were calculated using the results obtained with two available simulation meshes: a soft tissue “fine” mesh (with greater anatomical detail but longer simulation time) and a soft tissue “coarse” mesh (with poorer anatomical detail but shorter simulation time), giving a total of 18 error measures per patient. The number of mesh triangles and simulation time for each mesh and patient are described in Alcañiz et al. [15].

3D Slicer [16] was leveraged for the computation of the distance to the actual surgical outcome (previously registered to the preoperative image) in the two mentioned generated meshes, while ParaView [17] was chosen for taking the specific point measures. To do this, the sagittal plane and a horizontal plane at the height of the gonions were defined for each patient. These two planes were used for both distance meshes, fine and coarse, so that the corresponding points were very close (Figure 2).

Selecting anatomical landmarks manually leads to error, so for this study the interobserver error was also measured. For this purpose, distance values for both simulation meshes of one random patient were taken by two different observers and compared.

In addition, the mean of these 9 distances for each mesh and patient was considered (in order to have a representative value of the error of said patient mesh), as well as the absolute difference of these two mean distances for each patient (in order to measure the difference between the two types of mesh).

Finally, some variables obtained in our previous study [15] were also considered as follows: percentage of mesh vertexes with simulation error below 3 mm, both for the fine and the coarse mesh, and reduction percentage of simulation time between coarse mesh (few seconds) and fine mesh (mostly more than one minute). In total, 24 variables were collected per patient (for each type of mesh, 9 distances, 1 mean distance and 1 percentage of mesh vertexes with simulation error below 3 mm; then, 1 difference of mean distances and 1 reduction percentage of simulation time between coarse and fine meshes).

With the 24 variables defined, the following analyses were carried out:*Are the results of the fine mesh more accurate than those of the coarse mesh?* It was studied if there were significant differences between the distance in the fine mesh and that of the coarse mesh, in order to know if they are equivalent or not.*Are the results valid for clinical use?* It was studied if the distances were significantly smaller than 2 and 3 mm (the most common values in the literature for the clinically accepted error in orthognathic surgery), and therefore if the results are clinically valid.*Is the difference of mean distances between fine and coarse meshes correlated to simulation time?* The reduction percentage of simulation time from fine meshes to coarse meshes was compared with the difference of their mean distances, to understand if there is a correlation between these variables.*Is the mean distance of the anatomical landmarks equivalent to the one from mesh vertexes?* The mean distance of anatomical landmarks for each type of mesh was compared with its corresponding percentage of mesh vertexes with simulation error below 3 mm, to see if these measures are equivalent or not.

### 2.2. Qualitative Validation

The qualitative validation was composed by a series of questions asked to two surgeons regarding to the simulation results obtained for the 10 patients. These questions were set by two researchers, inspired by studies from the literature [9,11,12]. The aspects to evaluate in these questions were as follows: the anatomical detail of each mesh, the accuracy of the results (both in general and by specific regions), the texture and the influence of the simulations on the surgical planning process.

Some questions were specifically about the simulation results, and were therefore asked for each of the 10 patients in the study, while others were generic, and were asked only once to each surgeon. Two types of questions were considered: evaluation questions on a Likert scale (from 1 to 5) and binary questions (“yes” or “no”). They were defined in such a way that 5 and “yes” corresponded to the best result, while 1 and “no” corresponded to the worst result.

Specifically, for each patient, 12 evaluation questions and 7 binary questions were asked. In addition, at the end, 1 final evaluation question of the results and 10 generic binary questions were asked, for a total of 201 questions for each surgeon.

Questions of direct comparison between bone surgical planning for soft tissue simulation (taken from the postoperative image) and real bone preoperative planning (from the surgical planning program used by orthognathic surgeons at Hospital Universitario La Paz) were excluded, since generally they do not coincide exactly. This issue is due to surgical inaccuracies produced during the operation when trying to replicate the planning. In other words, surgeon error was eliminated and the actually performed surgery was considered as bone planning for soft tissue simulation.

The experiments performed for each patient are detailed in Appendix A and the corresponding questions are summarized in Table 1. An example of the soft tissue meshes from one patient shown in the experiments can be seen in Figure 3.

With the answers to the defined questions, the following analyses were performed:For evaluation questions, differences between the surgeons and differences for each surgeon before and after seeing the actual surgical result were studied, to see if they agreed with each other and if they changed their opinion.For binary questions, percentage of answer coincidence between the surgeons was calculated, to see their level of agreement.

### 2.3. Quantitative vs. Qualitative Comparison

Once both the quantitative and qualitative validations were completed, one last analysis was performed in order to answer the following question: *are quantitative errors correlated to qualitative evaluations?*

The qualitative results of the real general accuracy of the simulation (that is, the evaluation of both surgeons after seeing the actual surgical outcome) were compared with the quantitative results of the mean distance for each patient, to study whether these two types of validation are linearly related or not, and therefore are both necessary.

### 2.4. Statistical Methodology

All data was collected in Microsoft Excel and later analysed with the statistical program SAS 9.3 (SAS Institute, Cary, NC, USA).

For the aforementioned analyses, the following tests were used:The normality of the quantitative variables was studied using the Kolmogorov-Smirnov test. Qualitative evaluation questions were described with first quartile, median and third quartile.For paired data, the Student’s *t*-test (parametric variables) and the Wilcoxon signed rank test (non-parametric variables) were used. For unpaired data, Pearson’s correlation (parametric variables) and Spearman’s correlation (non-parametric variables) were used.

All statistical tests were considered bilateral, and a *p*-value of less than or equal to 0.05 was considered statistically significant.

## 3. Results

The results obtained for the three proposed analysis groups are presented below.

### 3.1. Quantitative Validation

Regarding the interobserver error, there were no significant differences between the mean distances measured by each observer, both for the fine mesh (*p* = 0.817) and for the coarse mesh (*p* = 0.527). The mean interobserver distance error was 0.3 mm for the anatomical points of the fine mesh and 0.75 mm for those of the coarse mesh, which are sub-millimetre errors. Although the coarse mesh mean interobserver distance error was greater than that for the fine mesh, there were no significant differences in their mean values (*p* = 0.106). Therefore, all image manipulation and error measurements for this study were performed by a single observer.

The normality test resulted in 8 variables that did not follow normality: A, LL, Pog, Me, LGo and mean for the fine mesh; Me for the coarse mesh; and the difference of mean distances. Therefore, in the description of the distances, the first quartile, median and third quartile were used for each anatomical landmark and type of mesh, for the mean distance per each type of mesh and for the difference of the mean distances (Table 2). Mean and standard deviation were used instead for the description of the percentage of mesh vertexes with simulation error below 3 mm for the fine mesh (94.9 ± 3.3%) and the coarse mesh (91.8 ± 4.4%), and the reduction percentage of simulation time between both meshes (90.3 ± 4.6 %).

The four proposed quantitative analyses gave the following results. When comparing the distances for the fine and the coarse meshes, there were no significant differences for Sn (*p* = 0.192), A (*p* = 0.114), UL (*p* = 0.285), LL (*p* = 0.959), B (*p* = 0.139), Pog (*p* = 0.445), RGo (*p* = 0.508), LGo (*p* = 0.241) and mean (*p* = 0.285). However, significant differences between the two meshes were observed for the Me distances (*p* = 0.011), which tend to increase in the coarse mesh.

Regarding the clinical use of these simulations, the previous results were compared with 2 mm and 3 mm (obtained *p*-values are shown in Table 3). For the fine and the coarse meshes, the distances for Sn, A, UL, B, Pog and Me, and the mean distance were significantly lower than 2 mm. RGo and LGo distances were not significantly lower than 2 mm (although the *p*-values were near the significance level), but were significantly lower than 3 mm. Lastly, LL distance was not significantly lower than 2 mm. For the fine mesh it was not significantly lower than 3 mm either (although the *p*-value almost reached the significance level), while for the coarse mesh it was significantly lower than 3 mm.

Regarding the relationship between the difference of mean distances between the two meshes and the reduction percentage of simulation time, there was no correlation between these two variables (*p* = 0.803).

Finally, when comparing the mean distance of the anatomical landmarks for each type of mesh with its corresponding percentage of mesh vertexes with simulation error below 3 mm, an inverse correlation (r = −0.612 and r = −0.468 for the fine and coarse meshes, respectively) was found, as expected, but the significance level was not reached (*p* = 0.060 and *p* = 0.172 for the fine and coarse meshes, respectively).

### 3.2. Qualitative Validation

The two qualitative analyses described previously, corresponding to the evaluation and binary questions, were performed.

For the evaluation questions, the first quartile, median and third quartile were computed (Table 4). Since the results of the evaluation of the anatomical detail of the coarse mesh were so low from the start, the rest of the experiments were carried out only with the simulation results of the fine mesh.

When comparing the two surgeons, significant differences were found for the following questions: “anatomical detail of fine mesh” (*p* = 0.047), “general accuracy (before)” (*p* = 0.047), “lips accuracy (before)” (*p* = 0.017), “general accuracy (after)” (*p* = 0.047) and “lips accuracy (after)” (*p* = 0.017). In the comparison of the general accuracy before and after seeing the actual result, no significant differences were obtained for surgeon A (*p* = 0.260), but they were found for surgeon B (*p* = 0.014). Moreover, grades of 4 and 4.5 were obtained from surgeons A and B respectively in the final evaluation question of the results.

For the binary questions, answers count for all patients of each surgeon and percentage of agreement between them is shown in Table 5. Moreover, an 80% of coincidence between the surgeons was obtained in the 10 generic binary questions.

### 3.3. Quantitative vs. Qualitative Comparison

Correlation between quantitative mean distance for the fine mesh and the qualitative evaluation for each surgeon after having seen the actual surgical outcome (from question “general accuracy (after)”) was studied, and a scatter plot was created with the 20 pairs of values (Figure 4). An inverse correlation between them was obtained (r = −0.327), but the significance level was not reached (*p* = 0.159).

## 4. Discussion

In this section, the obtained results are discussed in the same order in which they have been presented in the previous section, and they are compared with those of other similar studies from the literature.

### 4.1. Quantitative Validation

When comparing the distances obtained in the fine mesh with those corresponding to the coarse mesh, it is observed that there are not significant differences except for one anatomical landmark (Me), which may indicate the need for greater refinement of the soft tissue simulation mesh in the neck region. Even so, in general the two meshes are equivalent.

In relation to the clinical use of the results, all distances for both meshes and their mean distances are significantly lower than 2 mm, except those corresponding to LL, RGo and LGo. The gonions show *p*-values near the significance level, while LL presents higher *p*-values. When the tests are repeated with a value of 3 mm instead of 2 mm, distances for all anatomical landmarks are significantly smaller than 3 mm, except for LL of the fine mesh (which is anyway very close to the significance level), which is probably due to the different variable distributions and statistical tests performed. These results are consistent with those obtained in many studies in the literature, where greater simulation errors are described in the lower lip compared to other regions [4,6,8,12,14,18,19,20,21,22,23,24,25]. Once again, fine and coarse meshes are shown to be quantitatively equally valid for soft tissue simulation.

Regarding the correlation between the reduction percentage of simulation time and the difference of mean distances, these variables are not linearly related, and therefore higher simulation time reduction does not correspond to a greater simulation error. In other words, the simulation error is preserved, regardless of the simulation time, which means that faster simulations can be performed without the risk of suffering greater error in the result. This would again justify the use of the coarse meshes for the first steps of the preoperative planning.

Finally, the inverse correlation between the mean distance of anatomical landmarks for both meshes and its corresponding percentage of mesh vertexes with simulation error below 3 mm is reasonable. This relation is more appreciable for the fine mesh, where results could be more reliable. However, the significance level was not reached, which means that both variables should be considered separately when validating software simulation results. In the validation studies reviewed in the literature, only one of these variables is normally used: specific anatomical points of interest [19,21,26,27,28], regions defined with different landmarks and planes [29,30,31,32,33] or the whole face [10,11,13,15,20,22,34,35,36,37,38]. Very few cases consider both specific regions and the whole face [6,7,8,12,14,25].

### 4.2. Qualitative Validation

In this section, results of the evaluation questions (considering the median) and binary questions are analysed together, comparing the answers between the surgeons and, in those carried out before and after seeing the actual result, of the same surgeon.

Anatomical detail of the meshes: both surgeons consider that the fine mesh resembles the patient real anatomy (A = 4.5 and B = 4, due to lack of detail in the lips), although there is a significant difference between them (*p* = 0.047). However, both consider that the coarse mesh does not resemble the real anatomy (A = 2 and B = 1.5, since the characteristic features of the patient are not perceptible), with no significant differences. This means that, although the error is preserved in the coarse mesh and therefore it passes the quantitative validation, as mentioned previously, the surgeons do not consider this mesh realistic enough, so it does not pass the qualitative validation. In this regard, in the generic binary questions, surgeons comment that they could wait 1 min on average to obtain the final simulation result with the higher resolution mesh, but they do not agree on the use of the lower resolution mesh to carry out intermediate surgical planning steps in a few seconds (one surgeon would use it, the other would not).General accuracy: both surgeons consider that the simulation is a correct prediction of the possible result of the surgery (A = 4 and B = 4.5) and of the actual result (A = 4.25 and B = 4.75), even though there are significant differences between them in both cases (*p* = 0.047). The surgeons show a general positive opinion of the simulation results, although at different levels. In addition, both surgeons tend to increase the grade of the simulation accuracy when seeing the real result, as already stated in the literature [11]: for surgeon A (before = 4 and after = 4.25) the difference is not significant (*p* = 0.260), while for surgeon B (before = 4.5 and after = 4.75) it is (*p* = 0.014). Finally, the surgeons final scores to the simulations (A = 4 and B = 4.5, from the final evaluation question) coincide with these results, indicating that the overall experience matches the median of the evaluations.Regions accuracy: surgeons consider, both before and after seeing the actual result, that the simulation is correct, specifically, in the regions of the nose (A = 4.75 and B = 4.5 before, and A = B = 4.75 after) and chin (A = B = 4.75 before, and A = B = 5 after), without significant differences. However, both before and after seeing the actual result, there are significant differences between them (*p* = 0.017) in the evaluation of lips accuracy (A = 3.75 and B = 4.5 before, and A = 4 and B = 4.75 after), which is still high but lower than that of the other regions. This coincides with the quantitative results (where LL was one of the least accurate anatomical landmarks), but the difference between the two surgeons is appreciable, which shows different levels of satisfaction in this region. As in the general accuracy evaluation, the tendency of both surgeons when seeing the actual result is to maintain or increase the grade for the region accuracy of the nose (before = after = 4.75 for surgeon A, before = 4.5 and after = 4.75 for surgeon B), the lips (before = 3.75 and after = 4 for surgeon A, before = 4.5 and after = 4.75 for surgeon B) and the chin (before = 4.75 and after = 5 for both surgeons).Clinical use: before seeing the actual result, the simulation appears to be clinically acceptable in about half of the cases (surgeon A = 50% and surgeon B = 40%), and, after seeing it, in more than half of the cases (surgeon A = 60% and surgeon B = 80%). Even if these results are acceptable, they are way poorer than the quantitative results, where only two patients (patients 5 and 6) show a mean distance above 1 mm (Figure 4).Texture: both surgeons consider the textured simulation to be very realistic (surgeon A = 4.5 and surgeon B = 5), with no significant differences. In addition, they indicate that the simulation appearance improves by adding the texture in most cases (60% agreement) and therefore they would show the simulation to the patient (100% agreement), although they would need to manually modify some areas of the texture in most cases (100% agreement).Comparison with commercial software: the simulation seems much more reliable than the possible result from Dolphin Imaging (Dolphin Imaging & Management Solutions, Chatsworth, CA, USA), which is the commercial software used at Hospital Universitario La Paz for orthognathic surgery planning (surgeon A = 90% and surgeon B = 100%).Influence on the surgical planning process: both surgeons consider that in general the simulation would have influenced them considerably in the surgical planning process (A = 4.25 and B = 5, without significant differences), both to modify the bone planning if they were not satisfied with it and to maintain it if they were satisfied (90% agreement). This indicates the important role that surgical planning programs play for the surgeons, which show a great capacity to influence them.

On the one hand, evaluations are similar when comparing between surgeons, with differences especially in the region of the lips. In addition, in all the patient binary questions, at least 60% of agreement (2 out of 7 questions) between surgeons is achieved, even reaching 90% (2 out of 7) and 100% (2 out of 7). In the final questions, they indicate, among other things, that they like the results in general, they would use this planning tool in their daily work and they would trust these results more than those of commercial software, with an 80% of agreement. It can be therefore confirmed that there is concordance between them, but there are still some discrepancies that confirm the need to validate the simulation results with a larger group of experts.

On the other hand, the improvement of the evaluations after seeing the actual surgical results confirms the usefulness of this simulation tool. In other words, in several cases the surgeons thought that the real outcome would be different than the simulation result, which showed some initial mistrust. Once the resemblance between them was proved, the surgeons’ degree of confidence in the tool increased, and so did their willingness to use it in their daily work.

### 4.3. Quantitative vs. Qualitative Comparison

Finally, the mean distance of anatomical landmarks and the surgeon’s final evaluation for each patient (from question “general accuracy (after)”) are not correlated. Some patients show great errors (patient 6 = 1.52 mm) but also high evaluations (surgeon B = 4.5), as well as other patients show small errors (patient 7 = 0.72 mm) and low evaluations (surgeon A = 3). This experiment demonstrates that these variables are not linearly related and therefore both are necessary for a complete validation of the simulation results.

Very few works in the literature address both types of validation, and none analyse them statistically to justify the need to use both:In one of the first validation studies of soft tissue simulation results [10], the authors simulated 3 patient cases using FEM. Quantitatively, they obtained a mean error of the whole face between 1 and 1.5 mm, while for qualitative validation they made a visual comparison, first superimposing the meshes and then placing them side by side. They neither indicate the computation time nor texture the results.The study from Mollemans and colleagues [11] represents one of the most comprehensive simulation and validation works to date. The authors compared FEM, mass spring model (MSM) and mass tensor model (MTM) simulation methods in 10 patients, obtaining a 90th percentile error of 1.51 mm for FEM, with their own system to measure distances between the two meshes. An average time of 25.7 s is reported for the FEM simulation, and texture was applied to results for greater realism. Qualitative validation was carried out with 8 surgeons through 2 experiments, corresponding to questions before seeing the actual surgical outcome and after seeing it, as in our study. The means of the surgeons’ answers to 3 questions for each patient and another 3 generic ones are mostly positive. Finally, the authors briefly comment, without analysing them in detail, some inconsistencies between the qualitative and quantitative results, which justify the need to use both validations.In a recent publication [13], the authors performed an automatic segmentation, MSM simulation and surgical navigation study in a single patient. On the one hand, the quantitative validation was based on the computation of the distance with three different methods, obtaining a 91% of mesh error below 2 mm and a mean error below 1 mm. The qualitative validation, on the other hand, was based on 17 evaluation questions on a 4-point Likert scale, with 12 surgeons, but the questions and answers are not indicated. Moreover, the simulation result has no texture, and the computation time is not indicated either.Lastly, Kim D. et al. have recently published many studies on soft tissue simulation in orthognathic surgery, focusing mainly on the lip region. In 2017 they published a simulation and validation study [12] with a cohort of 40 patients, with both types of validation. For the quantitative part, they divided the face into 8 regions, and found that all of them, except the one corresponding to the lower lip, had a mean error below 1.5 mm and a maximum error below 3 mm. For the qualitative validation, they asked 2 surgeons if the simulation results were clinically acceptable, obtaining 32 out of 40 positive answers, but they did not relate these results to the quantitative ones. In their latest study [14], the authors simulated and validated the lip region in 35 patient cases. For the quantitative validation, they divided the face into 6 regions, obtaining a mean error of 1 mm. For the qualitative part, in this case they did not consult the surgeons but mathematically analysed the shape of the lips instead, assuming that it represents the opinion of a surgeon. They analysed only the lips, obtaining 26 out of 35 clinically acceptable results. In both publications, they textured the simulation result and highlighted that the main limitation of their study was the computation time, which was less than 10 min in their first study and approximately 30 min (due to the greater complexity of the algorithm) in their second study, which precludes its use in clinical practice.

### 4.4. Study Limitations and Sources of Error

In the current study, a quantitative and qualitative validation of a FEM-based pipeline for orthognathic surgery planning [15] that combines accuracy and computation time, as well as realism by adding the texture of the patient, is carried out. However, there are some limitations that should be considered for future studies. As described by Khambay and Ullah [7], some recommendations should be followed for a study of these characteristics to be valid. Some of them have already been mentioned, and most of them are fulfilled in this study: using the actual outcome (and not the surgical planning) to compare the results; registering the images considering the bone regions that are not modified; measuring the simulation error in absolute values, taking more than one type of measurement, specifically by regions and only in those where there is soft tissue change after the surgery. However, in the current study, the distance between the simulation mesh and the one from the actual postoperative result is determined by the distance to the closest point (using an iterative closest point algorithm), and not to the corresponding anatomical landmark, which involves an underestimation of the error [6]. Also, measuring errors by means of using points is a waste of the great potential of 3D technologies for these applications, so curves are recommended instead.

It is also important to consider the sources of error that greatly influence these types of analysis [24]. One of them is the variability in the identification of anatomical landmarks. In the present study, the interobserver error for one patient was measured to give an idea of its magnitude, but intraobserver error could also occur. The current trend is towards the automatic identification of landmarks, for example by means of artificial intelligence algorithms, which allow minimizing this error as well as saving time for surgeons.

A potential error comes also from the underlying simulation model itself, which is variable among publications. The most common simulation model among research studies of soft tissue simulation software in orthognathic surgery is FEM [9,10,12,14,15,22,23,27,36,39], but some solutions are based on MSM [13,28] and MTM [35]. One study even compared these three simulation models [11]. Moreover, the best-known commercial orthognathic surgery planning programs have different simulation algorithms: Maxilim (Medicim NV, Mechelen, Belgium) is based on MTM [8,20,29,31], 3dMDvultus (3dMD, Atlanta, GA, USA) relies on MSM [7,19,30,38], Dolphin Imaging uses a morphing algorithm [26,32,33], while ProPlan (Materialise, Leuven, Belgium) simulation is based on the finite difference method [6,32]. These different ways of predicting the soft tissue behaviour in response to bone movement hinder the comparison of several validation studies.

Another possible source of error is given by the variability among patients. In fact, several factors influence the soft tissue response and therefore the accuracy of the prediction [18]: gender (women tissue seems to deform more than that of men), race (commercial programs are mostly validated with data from Caucasian patients) and type of surgery (for example, bimaxillary surgeries entail greater error than monomaxillary ones). The relationship among these mentioned factors (as well as others, like the patient’s age) and the simulation error is also studied in other papers [8,15].

Finally, the type of quantitative analysis performed will also influence the results [30]. Depending on the study, validation is carried out considering the face as a whole, by anatomical regions or by landmarks, as stated in the introduction, so results cannot be compared [40]. In addition, different clinically accepted levels of error are presented in the literature [5]: 0.5 mm [19], 1 mm [13,36,38], 2 mm [6,7,8,20,22,25,26,28,31,32,34,37] and 3 mm [12,15,29,30].

For all the aforementioned reasons, a great bias is observed in these studies, based on weak methodologies, therefore a standard protocol for future studies is necessary [5,40]. Finally, bigger cohorts should also be collected, and more surgeons should be involved for completely exhaustive studies.

## 5. Conclusions

A validation study, both quantitative and qualitative, of the results obtained in 10 patients using a soft tissue simulation pipeline for clinical use in orthognathic surgery planning was carried out. The quantitative results were mostly accurate, with greater errors corresponding to gonions and lower lip, as stated in the literature, while the qualitative results showed general positive feedback from the surgeons, who answered similarly for most questions and whose evaluations improved when seeing the actual outcome of the surgery for each patient. The need of both types of validation to evaluate the results was demonstrated, since a weak correlation was found between the mean distance and the surgeons’ final evaluation. In addition, some study limitations were highlighted, and the need for a standard protocol for future studies has been stressed, so that variability in these types of analysis is minimized. Finally, the opinions of the surgeons are useful for the software development, in order to make relevant modifications in the future, which would allow to improve the results.

## Figures and Tables

**Figure 1 jpm-12-01460-f001:**
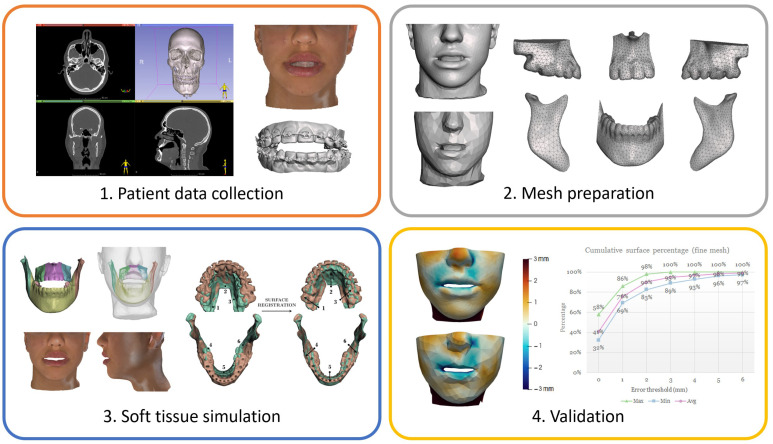
Scheme of the entire workflow presented in Alcañiz et al. [15] with the main steps: patient data collection (**left**: CBCT image; top right: textured face 3D scan; bottom right: dental 3D scan), mesh preparation (**top left**: soft tissue fine mesh; **bottom left**: soft tissue coarse mesh; **right**: bone fragments meshes), soft tissue simulation (**top left**: boundary conditions; **bottom left**: textured simulation output; **right**: preoperative [green] and postoperative [brown] 3D models from CBCT scans before [left] and after [right] registration) and first quantitative validation (**top left**: soft tissue simulation error for the fine mesh; **bottom left**: soft tissue simulation error for the coarse mesh; **right**: cumulative surface percentage plot for the fine mesh).

**Figure 2 jpm-12-01460-f002:**
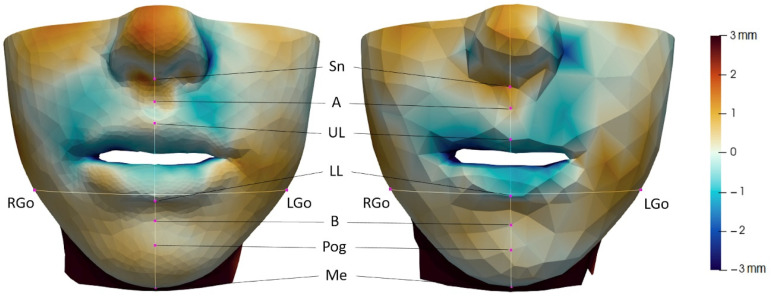
Anatomical landmarks where the distance to the actual surgical outcome was measured for the soft tissue fine mesh (**left**) and coarse mesh (**right**).

**Figure 3 jpm-12-01460-f003:**
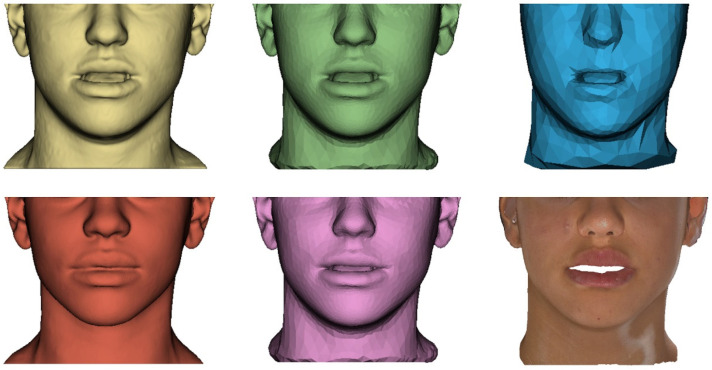
Example of the soft tissue meshes shown in the qualitative validation process for one patient. From left to right: in the first row, segmentation mesh from the preoperative CBCT image, fine mesh and coarse mesh; in the second row, segmentation mesh from the postoperative CBCT image, simulation result for the fine mesh and textured simulation result for the fine mesh.

**Figure 4 jpm-12-01460-f004:**
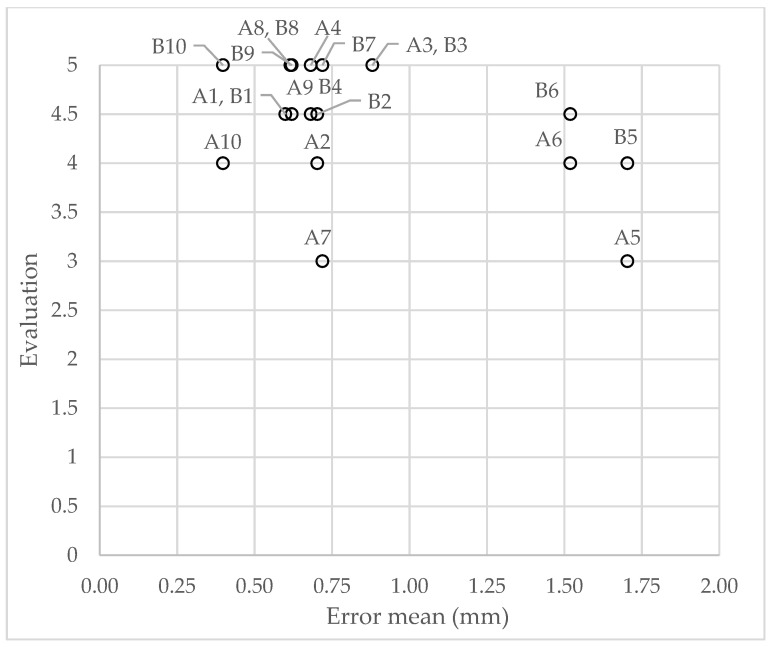
Scatter plot with the 20 pairs of values and labels corresponding to the mean distance for each patient (from patient 1 to 10) and its evaluation from each surgeon (A and B). A comma is used when evaluation from both surgeons coincided for a patient (for example A1, B1).

**Table 1 jpm-12-01460-t001:** Summary of the experiments and questions. In the columns: experiments performed, files shown during each experiment, questions asked to the surgeons and type of question (evaluation or binary). Some questions were asked both before showing the actual surgical outcome (before) and after showing it (after).

Experiment	Shown Files	Question	Type
Anatomical detail of the meshes	Segmentation mesh from the preoperative CBCT image and soft tissue fine and coarse meshes	Anatomical detail of fine mesh	Evaluation
		Anatomical detail of coarse mesh	Evaluation
Potential accuracy of the simulation	Preoperative CBCT image, bone planning and soft tissue simulation result	General accuracy (before)	Evaluation
		Nose accuracy (before)	Evaluation
		Lips accuracy (before)	Evaluation
		Chin accuracy (before)	Evaluation
		Valid for clinical use (before)	Binary
Real accuracy of the simulation	Postoperative CBCT image with its segmentation mesh and soft tissue simulation result	General accuracy (after)	Evaluation
		Nose accuracy (after)	Evaluation
		Lips accuracy (after)	Evaluation
		Chin accuracy (after)	Evaluation
		Valid for clinical use (after)	Binary
		Better than commercial software	Binary
Texture	Textured simulation result	Texture realism	Evaluation
		Improvement using texture	Binary
		Perfect texture	Binary
		Communication with patient	Binary
Influence on the surgical planning process	All the above	General influence	Evaluation
		Influence on the bone planning	Binary

**Table 2 jpm-12-01460-t002:** First quartile, median and third quartile of distances for each anatomical landmark and type of mesh, for the mean distance per each type of mesh and for the difference of the mean distances.

Variable	25th (mm)	Median (mm)	75th (mm)
Sn fine	0.26	0.99	1.35
Sn coarse	0.42	1.20	1.52
A fine	0.12	0.15	0.77
A coarse	0.31	0.50	0.97
UL fine	0.26	0.70	1.08
UL coarse	0.34	0.88	1.58
LL fine	0.31	0.92	2.03
LL coarse	0.35	0.73	2.27
B fine	0.15	0.45	0.75
B coarse	0.21	0.25	0.53
Pog fine	0.25	0.29	0.41
Pog coarse	0.20	0.27	0.51
Me fine	0.26	0.43	0.84
Me coarse	0.40	0.79	1.10
RGo fine	0.21	1.04	2.50
RGo coarse	0.36	1.57	2.13
LGo fine	0.38	0.63	2.70
LGo coarse	0.51	0.91	2.51
Mean fine	0.61	0.69	1.04
Mean coarse	0.69	0.84	1.30
Difference of mean distances	0.10	0.11	0.20

**Table 3 jpm-12-01460-t003:** *p*-values from the comparison of the distances of the anatomical landmarks and the mean distance for both meshes with 2 mm. For those that did not reach significance level (LL, RGo and LGo), comparison was also made with 3 mm.

Variable	Comparison Value (mm)	Fine Mesh	Coarse Mesh
Sn	2	<0.001	0.001
A	2	0.005	0.001
UL	2	<0.001	<0.001
LL	2	0.203	0.381
	3	0.059	0.023
B	2	<0.001	<0.001
Pog	2	0.005	<0.001
Me	2	0.007	0.017
RGo	2	0.073	0.074
	3	0.001	<0.001
LGo	2	0.074	0.162
	3	0.013	0.002
Mean	2	0.005	<0.001

**Table 4 jpm-12-01460-t004:** First quartile, median and third quartile of the answers to the evaluation questions for both surgeons (A and B).

Question	A			B		
	25th	Median	75th	25th	Median	75th
Anatomical detail of fine mesh	4.375	4.500	5.000	3.875	4.000	4.500
Anatomical detail of coarse mesh	2.000	2.000	3.000	1.375	1.500	2.125
General accuracy (before)	3.500	4.000	4.125	4.000	4.500	4.625
Nose accuracy (before)	3.750	4.750	5.000	4.000	4.500	5.000
Lips accuracy (before)	3.375	3.750	4.000	3.875	4.500	4.625
Chin accuracy (before)	4.000	4.750	5.000	4.500	4.750	5.000
General accuracy (after)	3.750	4.250	5.000	4.500	4.750	5.000
Nose accuracy (after)	4.000	4.750	5.000	4.375	4.750	5.000
Lips accuracy (after)	3.000	4.000	4.625	4.500	4.750	5.000
Chin accuracy (after)	4.000	5.000	5.000	5.000	5.000	5.000
Texture realism	4.250	4.500	5.000	4.500	5.000	5.000
General influence	3.750	4.250	5.000	4.375	5.000	5.000

**Table 5 jpm-12-01460-t005:** Summary of binary questions, answers count for all patients of each surgeon (A and B) and percentage of agreement between them.

Question	A		B		Agreement (%)
	Yes	No	Yes	No	
Valid for clinical use (before)	5	5	4	6	70
Valid for clinical use (after)	6	4	8	2	60
Better than commercial software	9	1	10	0	90
Improvement using texture	3	2	5	0	60
Perfect texture	1	4	1	4	100
Communication with patient	5	0	5	0	100
Influence on the bone planning	9	1	10	0	90

## Data Availability

Project data is available under request from the authors.

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
