# Peer review of "A Quantitative and Qualitative Clinical Validation of Soft Tissue Simulation for Orthognathic Surgery Planning"

_jpm, 2022, doi:10.3390/jpm12091460_

Round 1

Reviewer 1 Report

The total number of patients included in the study is too small. This is the part that is understandable enough. However, in the case of qualitative analysis, it is recommended to reinforce the results through the participation of more surgeons than two.

Reviewer 2 Report

Thank you for giving me this opportunity to review the article entitled, "A quantitative and qualitative clinical validation of soft tissue simulation for orthognathic surgery planning".

I carefully reviewed the submitted set of the manuscript and found it merits of publication. However:

- the entire manuscript needs a general formatting, following the rules of this journal (word "title", author's name; author contributions,...);

- Line 80:  The authors refer the number of participants, but not mention how they reached this number. Was it referring to the bibliography presented or was a sample calculation tool used?

- Methods section is too long; some information can go to the attachments; 

- Table 1: the header has to appear on the various pages of the table;

- Qualitative validation (in discussion): since there are only two surgeons in the qualitative validation, is the sample sufficient to discuss the topics presented in the discussion?

- Line 389: Dolphin Imaging - trademark symbol is missing.

- Line 417 - 453: this information can be used in the discussion but should be restructured, as it only describes the articles mentioned. I think you can adapt the information to the text and make it easier to read. It makes the discussion too big and tiring for the reader. 
